# The *NFKB1* Promoter Polymorphism (-94ins/delATTG) Is Associated with Susceptibility to Cytomegalovirus Infection after Kidney Transplantation and Should Have Implications on CMV Prophylaxis Regimens

**DOI:** 10.3390/cells10020380

**Published:** 2021-02-12

**Authors:** Hartmuth Nowak, Svenja Vornweg, Katharina Rump, Tim Rahmel, Matthias Unterberg, Björn Koos, Peter Schenker, Richard Viebahn, Michael Adamzik, Lars Bergmann

**Affiliations:** 1Department of Anesthesiology, Intensive Care Medicine and Pain Therapy, University Hospital Knappschaftskrankenhaus Bochum, Ruhr-University Bochum, In der Schornau 23-25, 44892 Bochum, Germany; svenja.vornweg2@kk-bochum.de (S.V.); katharina.k.rump@ruhr-uni-bochum.de (K.R.); tim.rahmel@ruhr-uni-bochum.de (T.R.); matthias.unterberg@kk-bochum.de (M.U.); bjoern.koos@ruhr-uni-bochum.de (B.K.); michael.adamzik@kk-bochum.de (M.A.); lars.bergmann@kk-bochum.de (L.B.); 2Department of Surgery, University Hospital Knappschaftskrankenhaus Bochum, Ruhr-University Bochum, In der Schornau 23-25, 44892 Bochum, Germany; peter.schenker@kk-bochum.de (P.S.); richard.viebahn@kk-bochum.de (R.V.)

**Keywords:** Cytomegalovirus, CMV, kidney transplantation, NFKB1 promotor polymorphism, non-coding DNA regions

## Abstract

Infections with cytomegalovirus (CMV) are one of the most frequent opportunistic infections in kidney transplant recipients. Current risk-adapted CMV chemoprophylaxis regimens are based almost solely on the donor and recipient CMV serostatus. Of note, the *NFKB1* -94ins/delATTG promoter polymorphism was recently associated with a higher risk of CMV infection. Since single genetic association studies suffer from poor reliability for drawing therapeutic implications, we performed this confirmatory study and included 256 kidney transplant recipients from 2007 to 2014 in this retrospective study. Patients were genotyped for the -94ins/delATTG *NFKB1* promoter polymorphism and followed up for 12 months. The incidence of CMV infection within 12 months after kidney transplantation was 37.5% (33/88) for the ins/ins, 21.5% (28/130) for the ins/del, and 23.7% (9/38) for the del/del genotypes (*p* = 0.023). Moreover, we evaluated the time of CMV infection onset. Ins/ins carriers had primarily late-onset CMV infection (median 194 days; interquartile range (IQR) 117–267 days) compared with heterozygous (ins/del; median 158 days; IQR 82–195 days) and homozygous deletion allele carriers (del/del; median 95 days; 84–123 days). Multivariate-restricted Cox regression model confirmed the ins/ins genotype to be an independent risk factor for the development of late-onset CMV infections. These findings should have an impact on post-kidney transplantation CMV chemoprophylaxis regimens.

## 1. Introduction

Infections with cytomegalovirus (CMV) are one of the most common opportunistic infections after kidney transplantation [1], with an impact on graft function and transplant rejection [2]. Moreover, they may trigger harmful CMV-associated diseases and might also influence mortality rates [3]. The incidence of CMV infection is strongly associated with the serostatus of donor and recipient in organ transplantation, whereas serostatus negative recipients (R^–^) have the highest risk when the donor is positive for CMV (D^+^) [4]. Therefore, in clinical context, different risk categories are distinguished by the CMV serostatus, leading to adapted anti-CMV strategies that incorporate antiviral chemoprophylaxis, and preemptive measures, whereas universal prophylactic regimens may harm due to drug toxicity, late CMV disease, and development of ganciclovir-resistant mutants [5,6,7,8]. Hence, differentiated risk stratifications are the cornerstone of modern antiviral chemoprophylaxis. However, the risk of CMV infection should not be attributed solely to the single risk factor of donor and recipient CMV serostatus, as genetic variation might also impact variability [9,10]. In particular, late-onset CMV infections after cessation of chemoprophylaxis have a more severe outcome [11]. Accordingly, there is a strong need for identifying new risk factors that can predict the development and onset of CMV infection. These will help to individualize the management of this infective complication, especially for a decision between prophylactic or preemptive strategies and its duration.

An interesting candidate gene for investigation of the impact of genetic variation on the risk of CMV infection is the nuclear transcription factor κB (*NF-κB*; nuclear factor kappa-light-chain enhancer of activated B cells). *NF-κB* is a family of dimeric transcription factors forming a regulative network that is known to amplify and perpetuate the inflammatory host response to infection, and to coordinate innate and adaptive immunity, cellular differentiation, proliferation, and survival [12]. Since *NF-κB* signaling stimulates transcription from the major immediate-early promoter (MIEP), it can also enhance CMV replication [13]. It comprises five protein monomers: *p65/RelA*, *RelB*, *c-Rel*, *p50*, and *p52* [14]. The mature *NF-κB* subunits *p50* and *p52* are processed out of the large precursors *p105* and *p100*, which are encoded by the genes *NFKB1* and *NFKB2* [15].

A functional insertion-deletion polymorphism (rs28362491) in the promoter of *NFKB1* (-94ins/delATTG) has been associated with an increased risk of developing infectious, autoimmune, and other inflammatory diseases [16,17,18], as well as the predisposition to several cancer types [19]. Recently, heterozygous and homozygous deletion carriers were associated with a higher risk of developing CMV infection after kidney transplantation [20]. Hence, the *NFKB1* -94ins/delATTG promoter polymorphism seems to be a promising candidate supporting the prediction of CMV infection and enrichment of risk-adapted anti-CMV strategies in terms of precision medicine.

However, since single genetic association studies do not have a high reliability and are likely affected by positive outcome bias [21], conclusions about therapeutic implications are difficult to draw. Therefore, we performed this retrospective study for evaluating and confirming the association of the *NFKB1* -94ins/delATTG promoter polymorphism with the risk of CMV infection in kidney transplant patients in our much larger patient population. Moreover, we investigated whether there is an effect of this promotor polymorphism on the onset of CMV infection.

## 2. Materials and Methods

### 2.1. Patients

This study was reviewed and approved by the local ethics board of the Medical Faculty of Ruhr-University Bochum (Bochum, Germany; approval number 4870-13). For study inclusion, written informed consent was obtained from all participating patients. The study was performed in accordance with the Declaration of Helsinki, Good Clinical Practice guidelines, and local regulatory legislations. For this study, we enrolled all kidney or simultaneous kidney and pancreas transplant recipients, who were treated between 2007 and 2014 at the Department of Surgery of University Hospital Knappschaftskrankenhaus Bochum, Bochum, Germany.

### 2.2. Treatments

Patients were recruited to donate a buccal swab for DNA extraction and genotyping of the -94ins/delATTG *NFKB1* promoter polymorphism after transplantation. Demographic and clinical data were collected at the time of study inclusion and patients were observed for a follow-up period of 1 year. All patients received immunosuppressive induction therapy with inter alia anti-thymocyte globulin or interleukin-2 receptor antibodies. Immunosuppressive maintenance therapy was performed according to local standards, which included steroids, calcineurin inhibitors, mTOR inhibitors, and/or mycophenolate mofetil (MMF).

Assignment to different CMV risk groups was done for each patient according to the individual pretransplant CMV serostatus of the recipient (R; R^+^ = CMV positive, R^–^ = CMV negative) and donor (D; D^+^ = CMV positive, D^–^ = CMV negative, or D^?^ = CMV serostatus unknown). Perioperative and postoperative risk-adapted antiviral chemoprophylaxis against CMV was performed with either ganciclovir or valganciclovir. High-risk patients (D^+^/R^–^) received chemoprophylaxis for 6 months, medium-risk patients (D^+^/R^+^ or D^–^/R^+^) for 3 months, and patients with low risk (D^–^/R^–^) received just perioperative prophylaxis. Routine surveillance for viral reactivation or infection was done for inpatients by weekly determinations of CMV viremia based on whole blood samples via polymerase chain reaction (PCR). After hospital discharge, detection of CMV viremia was continued monthly and when clinically indicated. Additionally, all patients were screened for CMV infection at the 1 year follow-up examination after transplantation.

### 2.3. Clinical Definitions

CMV infection was defined as the detection of viral nucleic acids in accordance with the definition of Ljungman et al. [22]. CMV DNA was evaluated using a commercially available PCR assay (Roche Ampliprep Assay, Roche Molecular Diagnostics, Pleasanton, CA, USA) according to the manufacturer’s instructions with calibration to the World Health Organization International Standard for Human CMV [23]. CMV disease and related entities (e.g., CMV pneumonia and CMV syndrome) were defined as the presence of CMV in the blood based on a local assay plus the presence of compatible symptoms, as described by Ljungman et al. [22].

Delayed graft function was defined as the necessity for hemodialysis within the first week after transplantation. When rejection was suspected, a biopsy of the kidney graft was performed and graded according to the Banff classification [24]. Hence, rejection was defined by biopsy-proven acute rejection (BPAR) [25].

### 2.4. DNA Genotyping

DNA samples were isolated from buccal swabs using QIAamp DNA Mini Kit (QIAGEN, Hilden, Germany). For genotyping of the -94ins/delATTG *NFKB1* promoter polymorphism, pyrosequencing was used. A 200 bp PCR fragment was amplified using primer *NFKB1*_del/ins_f(5′-ATGGACCGCATGACTCTATCAG-3′) and biotinylated primer *NFKB1*_del/ins_BIO_r(5′-GGGGCGCGCGTTAGGCGG-3′). PCR was performed at an annealing temperature of 60 °C in a 50 µL reaction mixture applying a commercially available PCR master mix (Eppendorf, Hamburg, Germany). Pyrosequencing was done on a PSQ96 MA (Pyrosequencing, Uppsala, Sweden) PCR machine using sequencing primer *NFKB1*_del/ins_seq(5′-CGTTCCCCGACCAT-3′). For genotype confirmation, randomly chosen samples were reanalyzed using a different nucleotide injection order.

### 2.5. Statistical Analysis

Baseline characteristics (timepoint of transplantation) and outcomes were analyzed as follows: Continuous variables are presented as mean ± standard deviation (SD) for normally distributed and median and interquartile range (IQR; 25th and 75th percentiles) for not normally distributed variables, as appropriate. Categorical variables are expressed as frequency and percentage. Comparison of continuous variables between groups was performed using a parametric one-way analysis of variance (ANOVA) or a nonparametric Kruskal–Wallis test, respectively. Categorical variables were compared by Pearson’s chi-square or Fisher’s exact test. The distributions of the *NFKB1* promoter polymorphism (-94ins/delATTG) were tested for deviations from the Hardy–Weinberg equilibrium (exact two-sided *p*-value; significance value 0.05). Time of survival free from CMV infection was assessed by the Kaplan–Meier method stratified by *NFKB1* genotype. The log-rank test was used to evaluate the univariate relationship between the *NFKB1* promoter polymorphism (-94ins/delATTG) genotype and the time of survival free from CMV infection. For assessment of the joint effect of the *NFKB1* promoter polymorphism (-94ins/delATTG) and potential predictors on CMV-free survival, a proportional hazards model (Cox regression) with single predictors was evaluated. Afterward, a multivariate restricted model was assessed using only those predictors with a *p*-value of 0.05 or lower based on the single predictor comparison. The final model included the following predictors: *NFKB1* promotor polymorphism (-94ins/delATTG) and CMV risk status by donor/recipient CMV serostatus. *NFKB1* genotypes for survival analyses were either stratified by all possible three alleles or by merging heterozygous and homozygous deletion carriers, resulting in two genotype groups (ins/ins and ins/del–del/del). Confidence intervals (CI) were calculated with a coverage of 95%. A two-sided *p*-value of less than 0.05 was considered statistically significant. Statistical analyses and graphical presentations were done by using The R Project for Statistical Computing Version 4.0.2 (The R Foundation for Statistical Computing, Vienna, Austria) with the tidyverse (version 1.3.0), survival (version 3.1-12), and survminer (version 0.4.7) packages.

## 3. Results

We included 256 kidney transplant recipients in this study, whereas 29.0% (74/256) received simultaneous pancreas kidney transplantation. According to the -94ins/delATTG *NFKB1* promoter polymorphism in comparison to the expected distribution of the Hardy–Weinberg equilibrium, 88 patients (34.4%) had homozygous insertion/insertion (ins/ins; expected *n* = 91), 130 (50.8%) heterozygous insertion/deletion (ins/del; expected *n* = 123), and 38 (14.8%) had the homozygous deletion/deletion (del/del; expected *n* = 41) genotype. No deviation from the Hardy–Weinberg equilibrium could be observed (*p* = 0.436).

### 3.1. Patient Characteristics

The baseline characteristics of these patients, stratified by genotype, are presented in Table 1. The mean age of the recipients was 53.23 ± 12.32 years, the majority was male (64.5%; 165/256). Pretransplant donor and recipient CMV serostatus was distributed as follows: 56 patients (21.9%) had a high risk (D^+^/R^–^), 154 (60.2%) were in the medium-risk group (D^+^/R^+^ or D^–^/R^+^), and 42 recipients (16.4%) had a low risk (D^–^/R^–^) for CMV infection. In four cases, CMV risk status was unknown. Deviation from in-house standard CMV chemoprophylaxis duration was inter alia attributable to CMV-positive blood transfusions. Cases of ganciclovir resistant CMV strains were not detected among the study patients. The majority of recipients (80.5%; 206/256) presented two or more HLA mismatches. Delayed graft function was observed in 27.3% (70/256) of all cases. There were no significant differences between the ins/ins, ins/del, and del/del genotypes’ demographics, type of transplantation, donor age, cold ischemia time, delayed graft function, immunosuppressive regimen, pretransplant CMV serostatus of donor and recipient, and duration of CMV chemoprophylaxis.

### 3.2. Outcomes

The outcomes, grouped by genotype, are presented in Table 2. CMV infections were observed in 27.3% (70/256) of all patients (18 in the high-CMV-risk group (D^+^/R^–^), 46 in medium-risk group (D^+/–^/R^+^), and 5 in low-risk group (D^–^/R^–^)). For the main result, we could observe a significant difference in the incidence of CMV infection between the three genotypes, whereas 37.5% (33/88) of ins/ins carriers developed a CMV infection within 1 year after kidney transplantation, compared to 21.5% (28/130) and 23.7% (9/38) of ins/del and del/del carriers, respectively. These differences were independent of donor/recipient CMV serostatus and duration of CMV chemoprophylaxis. Moreover, patients with ins/ins genotype developed infections with CMV demonstrably later after a median of 194 days (IQR: 117–267 days) post-transplantation, than deletion allele carriers (ins/del: 158; 82–195 days and del/del: 95: 84–123 days).

In the Kaplan–Meier analysis of the three genotypes (Figure 1a), a significant difference could just not be observed (*p* = 0.053). Since we could only identify 38 del/del carriers with nine CMV infections in our patients, we consequently merged heterozygous and homozygous deletion allele carriers into one group. In Kaplan–Meier analysis of the resulting two groups (Figure 1b), patients with the ins/ins genotype were associated with a higher risk of CMV infection than patients with deletion alleles (*p* = 0.016).

Altogether, 20 of 70 patients (28.6%) with CMV infection developed CMV disease. There was no difference in the incidence of CMV disease according to *NFKB1* promoter polymorphism. Moreover, 83 patients (32.4%) had a BPAR, also without any difference between genotypes.

In univariable Cox regression analysis (Table 3), the ins/ins genotype was associated with a higher risk of CMV infection compared to deletion allele carriers. Moreover, CMV risk status according to donor/recipient serostatus was also identified to be a risk factor for the development of CMV infection within one year after kidney transplantation. In a multivariable restricted model (Table 4), homozygous insertion allele genotype (hazard ratio 1.65; 95% CI 1.03–2.66; *p* = 0.038) and a donor positive/recipient negative (D^+^/R^–^) CMV serostatus (hazard ratio 2.84; 95% CI 1.05–7.661; *p* = 0.040) were identified to be an independent risk factor for developing a CMV infection within one year after kidney transplantation.

## 4. Discussion

In accordance with previous findings by Leone et al. [20], we can confirm that in our larger study population, the ins/ins genotype of the -94ins/delATTG *NFKB1* promoter polymorphism was also associated with a higher risk of developing a CMV infection within the first year after kidney transplantation. Multivariable proportional hazards model showed that the *NFKB1* promoter polymorphism was an independent risk factor for CMV infection with a 0.61-fold lower risk (95%CI: 0.38–0.97) for heterozygous and homozygous deletion allele carriers. In addition, to our knowledge, we can describe for the first time that the *NFKB1* promoter polymorphism also exerts a time-dependent effect on susceptibility to CMV infection, since patients with the ins/ins genotype not only had more CMV infections but also occurred later than in the deletion allele carriers.

Since CMV infections still have a high impact on the morbidity and mortality of transplant recipients, risk-adapted anti-CMV chemoprophylaxis is a cornerstone of modern post-transplantation management [2]. Chemoprophylaxis is usually performed with the application of antiviral drugs—namely, valganciclovir [26], which is given for up to 6 months [27]. In each individual case, the need for chemoprophylaxis and its duration is determined by the serostatus of the organ donor and recipient. However, one major drawback is the late onset of CMV infection and disease after discontinuation of chemoprophylaxis [28], which also has a high impact on the patients’ outcomes [21]. These late-onset CMV infections can also be seen in our data with a median CMV infection onset time of 126 and 209 days in patients with a chemoprophylaxis duration of 3 and 6 months, respectively. Late-onset CMV infection is commonly observed in high-risk patients (D^+^/R^–^) [29]. Interestingly, in our study population, most cases of CMV infection with 32.1% (18/56) were not only observed in high-risk patients, but also with 29.9% (46/154) in a comparable proportion of medium-risk patients (D^+/–^/R^+^), whereas patients with 3 months of chemoprophylaxis comprehensibly developed a CMV infection earlier than patients with 6 months. This observation could also be made in the study of Leone et al. [20]. This issue points out that a risk-adapted anti-CMV chemoprophylaxis, which is only based on donor and recipient CMV serostatus, might be an area of improvement, as a remarkably high proportion of kidney transplant recipients develop CMV infection within one year after transplantation after cessation of antiviral chemoprophylaxis.

Regarding the underlying *NFKB1* genotype, homozygous insertion carriers had the highest risk for CMV infection in comparison to heterozygous and homozygous deletion carriers. Hence, we can confirm an association of the -94ins/delATTG *NFKB1* promoter polymorphism with the risk for CMV infection, in accordance with the findings of Leone and colleagues [20]. Moreover, it should be particularly emphasized that ins/ins genotype patients had also primarily a late onset of CMV infection. Since late-onset CMV infections have a more severe outcome [21], we can imagine that homozygous insertion allele carriers, also under consideration of donor and recipient serostatus, might need a significantly longer CMV chemoprophylaxis than 6 months. Therefore, we suggest reevaluating current CMV chemoprophylaxis regimens regarding this promoter polymorphism. Future research with prospective studies is urgently needed to elucidate this issue.

The apparently pivotal role of the -94ins/delATTG *NFKB1* promoter polymorphism is not surprising, since it is well known that *NF-κB* is highly involved in the pathogenesis of CMV infection [13]. Although activation of *NF-κB* is essential for the resolution of CMV infection, it is also promoting viral replication over *NF-κB*-binding sites, which are included in the promoter of its key replication elements, especially the major immediate-early promoter (MIEP) [30]. It is assumed that the virus uses *NF-κB* activation for its own transcriptional events, and therefore inhibition of *NF-κB* activation may be a possible target for blocking virus replication [31]. In detail, it was shown by Kowalik et al. that nuclear *NF-κB* activity is increased in fibroblasts infected with CMV [32]. Later on, DeMeritt et al. observed that *NF-κB* activation occurs in a biphasic course with an increase of activation right at the beginning of infection and a second increase 8–12 h after [30]. At later time points of infection, CMV can also inhibit *NF-κB* activation [33,34] to maintain a careful balancing act between activation and inhibition [13]. These mechanisms are essential to maintain NF-κB activation at levels that are advantageous for viral replication and to escape from the cell’s antiviral defense program to persist for the lifetime of the host.

For the *NFKB1* -94ins/delATTG promoter polymorphism, it was reported that alterations in the promoter region affect the ability to bind transcription factors. As a result, deletion allele carriers have higher intracellular *NF-κB* levels than patients with an ins/ins genotype [18,35], leading to a higher capability of inflammatory responses [36]. Consequently, lower *NF-κB* levels of ins/ins genotype may facilitate the development of CMV infection and reactivation, especially under the condition of immunosuppression in kidney transplant patients. Interestingly, the presence of only one deletion allele in ins/del carriers showed a profound effect on the rate of CMV infection, which also did not differ in comparison to homozygous deletion allele carriers. We speculate that in contrast to a gene dosage effect, a threshold effect of *NF-κB* expression is of relevance here.

This study has some limitations. At first, an unrecognized selection bias, which is present in many genetic association studies, cannot completely be excluded. Moreover, although our patients were treated by a standardized multimodal regimen, undetected confounding factors might have influenced our results because of complex treatment algorithms in kidney transplantation, which include different immunosuppressive drugs and therefore can lead to variable immune reactions against CMV infection. However, the single-center design of this study might also be an advantage, as it limits the number of treatment protocols that can be used for kidney transplant recipients. Furthermore, our study, with a few exceptions, was conducted in patients of European-Caucasian descent; thus, the findings cannot be simply generalized to subjects of other ancestries.

## 5. Conclusions

In conclusion, our findings could confirm the reported association of the *NFKB1* -94ins/delATTG promoter polymorphism with the risk of CMV infection after kidney transplantation. Moreover, in our study population, we could describe for the first time that this promotor polymorphism also exerts a time-dependent effect on susceptibility to CMV infection, since homozygous insertion allele carriers not only had more but also primarily late-onset CMV infections. *NF-κB* plays a pivotal role in the development of CMV infection after kidney transplantation, and consequently genetic alterations like the *NFKB1* -94ins/delATTG promoter polymorphism seem to have a clinical impact and should therefore be taken into consideration for risk-adapted CMV prophylaxis regimens regarding donor and recipient serostatus. However, further prospective studies are needed to elucidate this issue.

## Figures and Tables

**Figure 1 cells-10-00380-f001:**
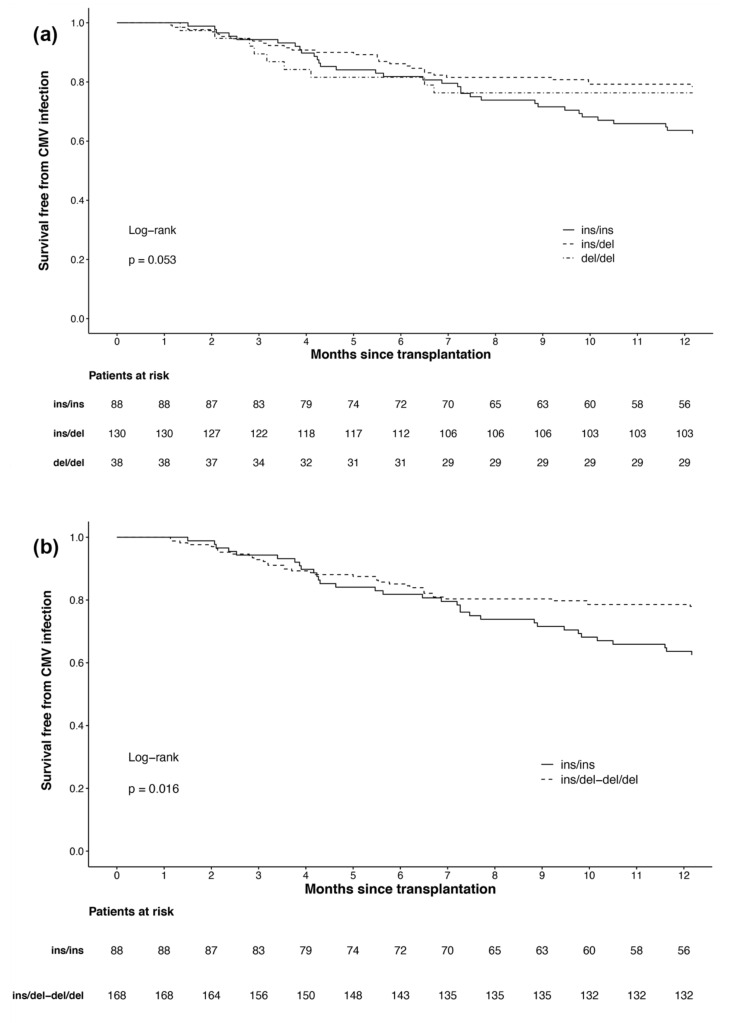
Survival free from cytomegalovirus infection in the first year after kidney transplantation by: (**a**) all three *NFKB1* genotypes; (**b**) *NFKB1* genotypes with merged groups for deletion allele carriers (ins/del and del/del).

**Table 1 cells-10-00380-t001:** Baseline characteristics of patients stratified by *NFKB1* genotype (*n* = 256).

Characteristic	ins ^1^/ins (*n* = 88)	ins/del ^2^ (*n* = 130)	del/del (*n* = 38)	*p*
Recipient age (y), mean ± SD	52.48 ± 11.51	53.09 ± 12.91	55.53 ± 12.16	0.452
Male sex, *n* (%)	59 (67.0)	80 (61.5)	26 (68.4)	0.606
Body mass index (kg/m^2^), mean ± SD	25.53 ± 3.72	25.87 ± 5.09	26.66 ± 3.78	0.466
Type of transplantation, *n* (%)				
Kidney	58 (65.9)	99 (76.2)	25 (65.8)	0.193
Combined kidney and pancreas	30 (34.1)	31 (23.8)	13 (34.2)	
Donor age (y), mean ± SD	48.69 ± 16.24	51.95 ± 16.76	53.76 ± 18.95	0.220
Cold ischemia time for kidney (h), mean ± SD	11.35 ± 4.31	11.42 ± 5.40	11.16 ± 5.44	0.961
Delayed graft function, *n* (%)	24 (27.3)	37 (28.5)	9 (23.7)	0.887
HLA ^3^ mismatch, *n* (%)				
0–1	12 (13.6)	19 (14.6)	3 (7.9)	0.345
2–4	51 (58.0)	73 (56.2)	17 (44.7)	
≥ 5	21 (23.9)	31 (23.8)	13 (34.2)	
Unknown	4 (4.5)	7 (5.4)	5 (13.2)	
Induction with ATG ^4^, *n* (%)	77 (87.50)	107 (82.31)	32 (84.21)	0.584
Immunosuppressive regimen, *n* (%)				
MMF ^5^, prednisone and tacrolimus	70 (79.5)	94 (72.3)	32 (84.2)	0.194
MMF, prednisone and cyclosporine	8 (9.1)	11 (8.5)	0 (0.00)	
Other	10 (11.4)	25 (19.2)	6 (15.8)	
Pretransplant CMV ^6^ donor (D)/recipient (R) serostatus, *n* (%)				
High risk (D^+^/R^–^) ^7^	18 (20.5)	29 (22.3)	9 (23.7)	0.294
Medium risk (D^+/–^/R^+^) ^8^	61 (69.3)	72 (55.4)	21 (55.3)	
Low risk (D^–^/R^–^) ^9^	9 (10.2)	26 (20.0)	7 (18.4)	
Unknown	0 (0.00)	3 (2.3)	1 (2.6)	
Prophylactic anti-CMV therapy, *n* (%)				
Perioperative	5 (5.7)	21 (16.2)	3 (7.9)	0.251
3 months	55 (62.5)	77 (59.2)	24 (63.2)	
6 months	24 (27.3)	25 (19.2)	10 (26.3)	
Unknown	4 (4.5)	7 (5.4)	1 (2.6)	

^1^ Insertion allele. ^2^ Deletion allele. ^3^ Human leukocyte antigen. ^4^ Anti-thymocyte globulin. ^5^ Mycophenolate mofetil. ^6^ Cytomegalovirus. ^7^ Donor positive, recipient negative. ^8^ Donor positive or negative, recipient positive. ^9^ Donor negative, recipient negative. Missing data were excluded from the analysis: 4 cases are missing for recipient age, 6 cases are missing for body mass index, 4 cases are missing for cold ischemia time for kidney, 7 cases are missing for delayed graft function.

**Table 2 cells-10-00380-t002:** CMV infection, CMV disease, and acute kidney rejection of patients stratified by *NFKB1* genotype (*n* = 256).

Outcome	ins ^1^/ins (*n* = 88)	ins/del ^2^ (*n* = 130)	del/del (*n* = 38)	*p*
CMV ^3^ infection, *n* (%)	33 (37.5)	28 (21.5)	9 (23.7)	0.023
**Donor/Recipient CMV serostatus**				
D^+^/R^– 4^ (*n* = 18/56; 32.1%)	7 (21.2)	9 (32.1)	2 (22.2)	0.610
D^+/–^/R^+ 5^ (*n* = 46/154; 29.9%)	24 (72.7)	17 (60.7)	5 (55.6)	0.108
D^–^/R^– 6^ (*n* = 5/42; 11.9%)	2 (6.1)	2 (7.1)	1 (11.1)	0.418
Unknown (*n* = 1/4; 25.0%)	0 (0.0)	0 (0.0)	1 (11.1)	0.665
**Duration of CMV chemoprophylaxis**				
Perioperative (*n* = 5/29; 17.2%)	1 (3.0)	3 (10.7)	1 (11.1)	0.705
3 months (*n* = 38/156; 24.4%)	18 (54.5)	16 (57.1)	4 (44.4)	0.183
6 months (*n* = 25/59; 42.4%)	13 (39.4)	8 (28.6)	4 (44.4)	0.288
Unknown (*n* = 2/12; 16.7%)	1 (3.0)	1 (3.6)	0 (0.0)	0.269
Time of transplantation to CMV infection (d), median (IQR)	194 (117–267)	158 (82–195)	95 (84–123)	0.025
**Donor/Recipient CMV serostatus**				
D^+^/R^–^ (127; 98–194)	129 (122–290)	126 (106–173)	62 (51–73)	0.098
D^+/–^/R^+^ (166; 97–218)	200 (123–240)	165 (72–195)	106 (95–123)	0.114
D^–^/R^–^ (87; 45–184)	196 (121–272)	110 (72–147)	87 (87–87)	0.819
Unknown (195; 195–195)	–	–	195 (195–195)	1.000
**Duration of CMV chemoprophylaxis**				
Perioperative (64; 35–87)	139 (139–139)	35 (34–50)	87 (87–87)	0.202
3 months (126; 95–172)	128 (114–222)	138 (95–168)	90 (73–102)	0.093
6 months (209; 186–284)	231 (206–293)	202 (161–282)	150 (95–196)	0.115
Unknown (172; 108–236)	45 (45–45)	87 (87–87)	–	0.317
CMV disease, *n* (%)	8 (9.1)	9 (6.9)	3 (7.9)	0.829
BPAR ^7^, *n* (%)	34 (38.6)	38 (29.2)	11 (28.9)	0.307

^1^ Insertion allele. ^2^ Deletion allele. ^3^ Cytomegalovirus. ^4^ Donor positive, recipient negative. ^5^ Donor positive or negative, recipient positive. ^6^ Donor negative, recipient negative. ^7^ Biopsy-proven acute rejection. Missing data were excluded from the analysis: 6 cases are missing for CMV infection, 6 cases are missing for CMV disease.

**Table 3 cells-10-00380-t003:** Univariable Cox regression analysis of kidney transplantation recipients according to risk of CMV infection.

	HR ^1^	95% CI ^2^	*p*
*NFKB1* promotor polymorphism (-94ins/delATTG)			
ins/ins ^3^	1	–	–
ins/del–del/del ^4^	0.568	0.355–0.908	0.018
Recipient age (per year)	0.999	0.980–1.018	0.905
Recipient sex male (vs. female)	1.002	0.614–1.634	0.994
Recipient body mass index (per 1)	1.021	0.968–1.077	0.448
Type of transplantation			
Kidney	1	–	–
Kidney + pancreas	1.065	0.639–1.775	0.810
Donor age (per year)	1.005	0.990–1.109	0.521
Cold ischemia time (per hour)	1.021	0.975–1.068	0.385
Delayed graft function (vs. none)	1.364	0.818–2.276	0.235
BPAR ^5^ (vs. none)	1.115	0.681–1.827	0.666
Induction with ATG ^6^ (vs. other)	1.127	0.577–2.202	0.726
Immunosuppressive regimen			
MMF ^7^, prednisone and tacrolimus	1	–	–
MMF, prednisone and cyclosporine	1.311	0.563–3.053	0.530
Other	1.138	0.608–2.132	0.686
CMV ^8^ risk status			
Low risk (D^–^/R^–^) ^9^	1	–	–
Medium risk (D^+/–^/R^+^) ^10^	2.724	1.082–6.858	0.033
High risk (D^+^/R^– ^)^11^	3.029	1.124–8.161	0.028
Prophylactic anti-CMV therapy			
Perioperative	1	–	–
3 months	1.409	0.555–3.580	0.471
6 months	2.484	0.951–6.492	0.063

^1^ Hazard ratio. ^2^ 95% Confidence interval. ^3^ Homozygous insertion carriers. ^4^ Merged group of heterozygous and homozygous deletion carriers. ^5^ Biopsy-proven acute rejection. ^6^ Anti-thymocyte globulin. ^7^ Mycophenolate mofetil. ^8^ Cytomegalovirus. ^9^ Donor positive, recipient negative. ^10^ Donor positive or negative, recipient positive. ^11^ Donor negative, recipient negative. Missing data were excluded from the analysis: 4 cases are missing for recipient age, 6 cases are missing for body mass index, 4 cases are missing for cold ischemia time for kidney transplantation, 7 cases are missing for delayed graft function, 4 cases are missing for CMV risk status, 12 cases are missing for prophylactic anti-CMV therapy.

**Table 4 cells-10-00380-t004:** Multivariable-restricted Cox regression analysis model of kidney transplantation recipients according to the risk of CMV infection.

	HR ^1^	95% CI ^2^	*p*
*NFKB1* promotor polymorphism (-94ins/delATTG)			
ins/ins ^3^	1	–	–
ins/del–del/del ^4^	0.605	0.376–0.973	0.038
CMV ^5^ risk status			
Low risk (D^–^/R^–^) ^6^	1	–	–
Medium risk (D^+/–^/R^+^) ^7^	2.468	0.975–6.247	0.057
High risk (D^+^/R^–^) ^8^	2.837	1.051–7.661	0.040

^1^ Hazard ratio. ^2^ 95% Confidence interval. ^3^ Homozygous insertion carriers. ^4^ Merged group of heterozygous and homozygous deletion carriers. ^5^ Cytomegalovirus. ^6^ Donor positive, recipient negative. ^7^ Donor positive or negative, recipient positive. ^8^ Donor negative, recipient negative. Missing data were excluded from the analysis: 4 cases are missing for multivariable restricted Cox regression.

## Data Availability

The data that support the findings of this study are available from the corresponding author, H.N., upon reasonable request.

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
