# Peer review of "The NFKB1 Promoter Polymorphism (-94ins/delATTG) Is Associated with Susceptibility to Cytomegalovirus Infection after Kidney Transplantation and Should Have Implications on CMV Prophylaxis Regimens"

_cells, 2021, doi:10.3390/cells10020380_

Round 1

Reviewer 1 Report

Authors present interesting correlations of NFKB1 -94ins/del ATTG promoter polymorphism and the risk of CMV infection after kidney transplantation. They find that this polymorphism is associated with a higher risk of developing CMV infection within the first year after kidney transplantation. The manuscript is written well and is easy to understand. This retrospective study is designed well and includes proper controls. My only comment is to extend the discussion on mechanism of NFkB regulation of CMV infection, specifically with regards to the timing of infection. It's been known that an early induction of NFkB may be detrimental to virus growth; however a late induction would help virus replication.

Author Response

Point 1: Authors present interesting correlations of NFKB1 -94ins/del ATTG promoter polymorphism and the risk of CMV infection after kidney transplantation. They find that this polymorphism is associated with a higher risk of developing CMV infection within the first year after kidney transplantation. The manuscript is written well and is easy to understand. This retrospective study is designed well and includes proper controls. My only comment is to extend the discussion on mechanism of NFkB regulation of CMV infection, specifically with regards to the timing of infection. It's been known that an early induction of NFkB may be detrimental to virus growth; however a late induction would help virus replication. 

Response 1: Thank you very much for your positive review of our study. Referring to your suggestions for broadening the discussion regarding the mechanisms of NFKb regulation of CMV infection, we have added the following lines: “At later time points of infection, CMV can also inhibit NF-κB activation [33,34] to maintain a careful balancing act between activation and inhibition [13]. These mechanisms are essential to maintain NF-κB activation at levels that are advantageous for viral replication and to escape from the cell’s antiviral defense program to persist for the lifetime of the host. For the NFKB1 -94ins/delATTG promoter polymorphism, it was reported that al-terations in the promoter region affect the ability to bind transcription factors. As a result, deletion allele carriers have higher intracellular NF-κB levels than patients with an ins/ins genotype [18,35], leading to a higher capability of inflammatory responses [36]. Con-sequently, lower NF-κB levels of ins/ins genotype may facilitate the development of CMV infection and reactivation, especially under the condition of immunosuppression in kidney transplant patients.” (lines 453-464)

Reviewer 2 Report

This manuscript describes a retrospective study comparing the CMV infection risk, and time to infection, of kidney transplant patients with and without a 4pb deletion in the promoter region of their NFKB1 gene.  The authors provide data that shows a significant association with a higher risk of CMV infection after kidney transplant for patients homozygous for the insertion, regardless of prior risk score. the authors conclude with the suggestion that NFKB1 genotype should be included when assessing patients for post-operative chemoprophylaxis regimens.

The scientific method and logic of the study is easy to follow, and the manuscript was a pleasure to read.  Extensive and appropriate statistical analyses, taking into account a variety of variables, were well-presented.

The manuscript does need some moderate English language editing (e.g. word choice and punctuation) to improve clarity and in some cases avoid misunderstanding of claims.  

I also suggest that only the first instance of the use of an abbreviation in a given table be given a superscript.

Author Response

Point 1: This manuscript describes a retrospective study comparing the CMV infection risk, and time to infection, of kidney transplant patients with and without a 4pb deletion in the promoter region of their NFKB1 gene.  The authors provide data that shows a significant association with a higher risk of CMV infection after kidney transplant for patients homozygous for the insertion, regardless of prior risk score. the authors conclude with the suggestion that NFKB1 genotype should be included when assessing patients for post-operative chemoprophylaxis regimens.

The scientific method and logic of the study is easy to follow, and the manuscript was a pleasure to read.  Extensive and appropriate statistical analyses, taking into account a variety of variables, were well-presented.

The manuscript does need some moderate English language editing (e.g. word choice and punctuation) to improve clarity and in some cases avoid misunderstanding of claims.

Response 1: Thank you very much for your positive review of our study. With regard to your comments on the English language, we have again critically reviewed the entire manuscript and made appropriate changes in order to improve clarity and to avoid misunderstanding of claims. All changes made are marked in the new version of our manuscript.

Point 2: I also suggest that only the first instance of the use of an abbreviation in a given table be given a superscript.

Response 2: Thank you very much for this suggestion. We accordingly changed the tables in the manuscript. Now only the first instance of the use of an abbreviation in a table is given a superscript.